# Anisotropic nanocrystal superlattices overcoming intrinsic light outcoupling efficiency limit in perovskite quantum dot light-emitting diodes

Sudhir Kumar [1,5], Tommaso Marcato [1,5], Frank Krumeich [2], Yen-Ting Li[3,4], Yu-Cheng Chiu [3] & Chih-Jen Shih [1✉]

Quantum dot (QD) light-emitting diodes (LEDs) are emerging as one of the most promising candidates for next-generation displays. However, their intrinsic light outcoupling efficiency remains considerably lower than the organic counterpart, because it is not yet possible to control the transition-dipole-moment (TDM) orientation in QD solids at device level. Here, using the colloidal lead halide perovskite anisotropic nanocrystals (ANCs) as a model system, we report a directed self-assembly approach to form the anisotropic nanocrystal superlattices (ANSLs). Emission polarization in individual ANCs rescales the radiation from horizontal and vertical transition dipoles, effectively resulting in preferentially horizontal TDM orientation. Based on the emissive thin films comprised of ANSLs, we demonstrate an enhanced ratio of horizontal dipole up to 0.75, enhancing the theoretical light outcoupling efficiency of greater than 30%. Our optimized single-junction QD LEDs showed peak external quantum efficiency of up to 24.96%, comparable to state-of-the-art organic LEDs.

[1] Institute for Chemical and Bioengineering, ETH Zürich, 8093 Zürich, Switzerland. [2] Laboratory of Inorganic Chemistry, ETH Zürich, 8093 Zürich, Switzerland. [3] Department of Chemical Engineering, National Taiwan University of Science and Technology, Taipei 10607, Taiwan, ROC. [4] National Synchrotron Radiation Research Center, Hsinchu 30076, Taiwan, ROC. [5] These authors contributed equally: Sudhir Kumar, Tommaso Marcato. ✉email: chih-jen.shih@chem.ethz.ch

Quantum dot (QD) light-emitting diodes (LEDs) are ideal for next-generation flat-panel displays because of their scalability, cost effectiveness, emission color purity, and tunable chromaticity[1–5]. Considerable research efforts in the past two decades have demonstrated high-efficiency QD LEDs using CdSe, InP, and lead halide perovskite (LHP) nanocrystals (NCs), with external quantum efficiencies, $\eta_{ext}$, of up to 20.5%[4], 21.4%[6], and 23.4%[7–9], respectively. To date, the methods used to enhance device performance have mainly focused on the passivation of NC surface defects. Taking the defect-tolerant LHP QD system[10–13] as an example, ligand engineering, modulation of stoichiometric compositions, and cation/anion mixing[14–18] have been employed to substantially improve the device performance[7–9]. Nevertheless, as the internal quantum efficiency has approached unity, new strategies that boost the intrinsic light outcoupling efficiency, $\eta_{out}$, become increasingly attractive to bring the device performance to the next level.

In thin-film LEDs, it is well-known that majority of radiation generated within the emissive layer is trapped inside the thin-film stack through the dissipating pathways of waveguide, surface plasmon, and substrate modes, eventually converting to heat. The classical ray optics theory gives a first approximation, $\eta_{out} = 1/(2n^2)$, where $n$ is the refractive index of emissive layer[19]. For example, in the LHP systems, given $n \sim 2.0–2.6$, one could estimate $\eta_{out}$ of only ~15%[20–22]. An approach to address this challenge is to intrinsically increase $\eta_{out}$ by inducing emission directionality in quantum emitters. For example, in organic LEDs (OLEDs), orienting emissive molecules with their transition dipole moments (TDMs) parallel to the substrate would outcouple more light within the critical angle for the glass-air interface[23–25]. The major advantage of this approach is to allow large-scale manufacturing without the need of other physical light-extraction techniques, such as microlens arrays, photonic crystals, surface corrugation, and matching index grating[26–28], which often demand expensive and complicated fabrication[28,29] and even compromise emission chromaticity from different viewing angles[28,30].

Spontaneous emission in organic molecules involves highly localized excitons, which offer clues to the correlation between TDM and molecular orientations that could guide rational molecular design and thin-film processing[25]. In contrast, excitons in inorganic semiconductor crystals reside in extended band edge states of complex three-dimensional (3D) symmetry, which usually leads to isotropic TDM orientation. An ideal candidate to control the TDM orientation for LEDs is the atomically thin two-dimensional (2D) materials, such as $MoS_2$ monolayer[31], because the 2D electronic structure confines all bright excitons in-plane[32]. The major challenge is that 2D monolayers are very sensitive to surface defects that drastically quench photoluminescence (PL) by orders of magnitude, particularly at high exciton concentrations, hindering practical applications[33].

Recent experimental studies have also suggested that excitons in the few-monolayer-thick colloidal CdSe nanoplatelets (NPLs) have completely in-plane TDM orientation[34–36]. Indeed, emission in the quantum-confined zinc-blende (ZB) crystal structure comes exclusively from the heavy-hole states, which accidentally have only mixed $p_x$ and $p_y$ symmetry, forming a bright plane that coincides with the platelet plane[34]. The CdSe NPLs, however, suffer from relatively low PL quantum yield, $\eta_{PL}$, and more critically, lose the preferential TDM orientation in their stacks[36]. To our knowledge, preferentially horizontal TDM orientation in QD-assembled thin films has never been demonstrated at device level.

With the above background in mind, to intrinsically enhance light outcoupling in QD-assembled thin films for high-efficiency LEDs, a challenging list of requirements must be met, including TDM orientation control in individual NCs, controlled NC assembly without compromising $\eta_{PL}$ and TDM orientation, and ligand engineering that balance dielectric confinement and carrier injection. In this report, we show that all these requirements can be satisfied by a scalable 2D superlattice system comprised of LHP ANCs.

## Results and discussions

**Emission polarization in individual ANCs.** First, we would like to point out that ANCs do not necessarily possess 2D electronic structure. Unlike the ZB crystal system, the LHP is of cubic structure with octahedral symmetry, where the bandgap occurs at the **R** point in the Brillouin zone, isomorphic to the $\mathbf{\Gamma}$ point. The spin-orbit interaction leads to the splitting of the conduction band states into a fourfold $\Gamma_8^-$ and a twofold $\Gamma_6^-$ states, and the valence band $\Gamma_6^+$ states are $s$-like having zero orbital angular momentum[37,38]. A proper description for spontaneous emission in LHP ANCs corresponds to the transition between the $\Gamma_6^-$ and $\Gamma_6^+$ states, which is of mixed $p_x$, $p_y$, and $p_z$ symmetry (Supplementary Section 1.1). We therefore deduce that, unless LHP ANCs are of atomic thickness that largely changes the band structure, the TDM orientation remains isotropic, in spite of a degree of quantum confinement.

It should be noted that controlling TDM orientation is not the only approach to induce emission directionality in individual QDs. Early findings from single-molecule spectroscopy have explored the effect of emission polarization in anisotropically dielectric-confined nanostructures[39,40]. Specifically, consider an isolated NC surrounded by low-dielectric-constant ligand medium, with the dielectric constants of $\epsilon_{NC}$ and $\epsilon_m$, respectively. The local electric field within NC, $\mathbf{E}^{loc}$, induced by an external field, $\mathbf{E}$, would strongly depend on the NC shape and the dielectric contrast, $\widetilde{\epsilon} = \epsilon_{NC}/\epsilon_m$, characterized by the local field factors, $f_i = E_i^{loc}/E_i$, where subscript $i$ correspond to $x$, $y$, and $z$ coordinates in space (Supplementary Section 1.2). Figure 1a presents a set of calculated ratios of local field factor, $f_x/f_z$, as a function of the NC aspect ratio, AR, for spheroids and square cuboids, in which $f_x = f_y$, using the dielectric constants in our synthesized LHP NCs ($\epsilon_{NC} = 4.7$ and $\epsilon_m = 2.129$)[34,41–43]. The cubic and spherical NCs (AR = 1) are characterized by their isotropic polarization response, which then departs from unity as the shape evolves towards asymptotic rod (AR → 0) and disk (AR → ∞). Specifically, the ANC has its in-plane (IP, $x$ or $y$ direction) local field factor increasing with AR, but the other way around for the out-of-plane (OP, $z$ direction) component. When AR → ∞, $f_x = 1$ and $f_z = 1/\widetilde{\epsilon}$ (see Supplementary Section 1.2), giving the theoretical upper limit for $f_x/f_z = \widetilde{\epsilon}$.

The emission radiative rate for a given dipole is proportional to $(\mathbf{p} \cdot \mathbf{f})^2$, where $\mathbf{p}$ and $\mathbf{f}$ are the transition dipole moment and local field factor vectors, respectively (Supplementary section 1.3). As a result, in a square ANC with $f_x = f_y$, the emission from an OP dipole is significantly screened and that from an IP dipole is amplified, resulting in an enhanced emission directionality toward the OP direction. Under the assumption of isotropic TDM orientation, the effective IP dipole ratio within individual NC determined by far-field measurement, $\Theta_{IP}$, is given by

$$\Theta_{IP} = \frac{2|f_x|^2}{2|f_x|^2 + |f_z|^2} \tag{1}$$

Accordingly, Eq. (1) and Fig. 1a reveal that a cubic NC with $f_x/f_z = 1$ would give $\Theta_{IP}$ of 0.67, corresponding to isotropic radiation, but the square LHP ANCs could yield $\Theta_{IP}$ up to 0.91 for AR → ∞ (Supplementary section 1.2), even with isotropic TDM orientation. We notice that the predicted $\Theta_{IP}$ upper limit nicely agrees with the experimentally measured values in 3-monolayer LHP ANCs (thickness $d = 1.8$ nm and AR ~ 30) in which $\Theta_{IP} = 0.81–0.85$ [44,45].

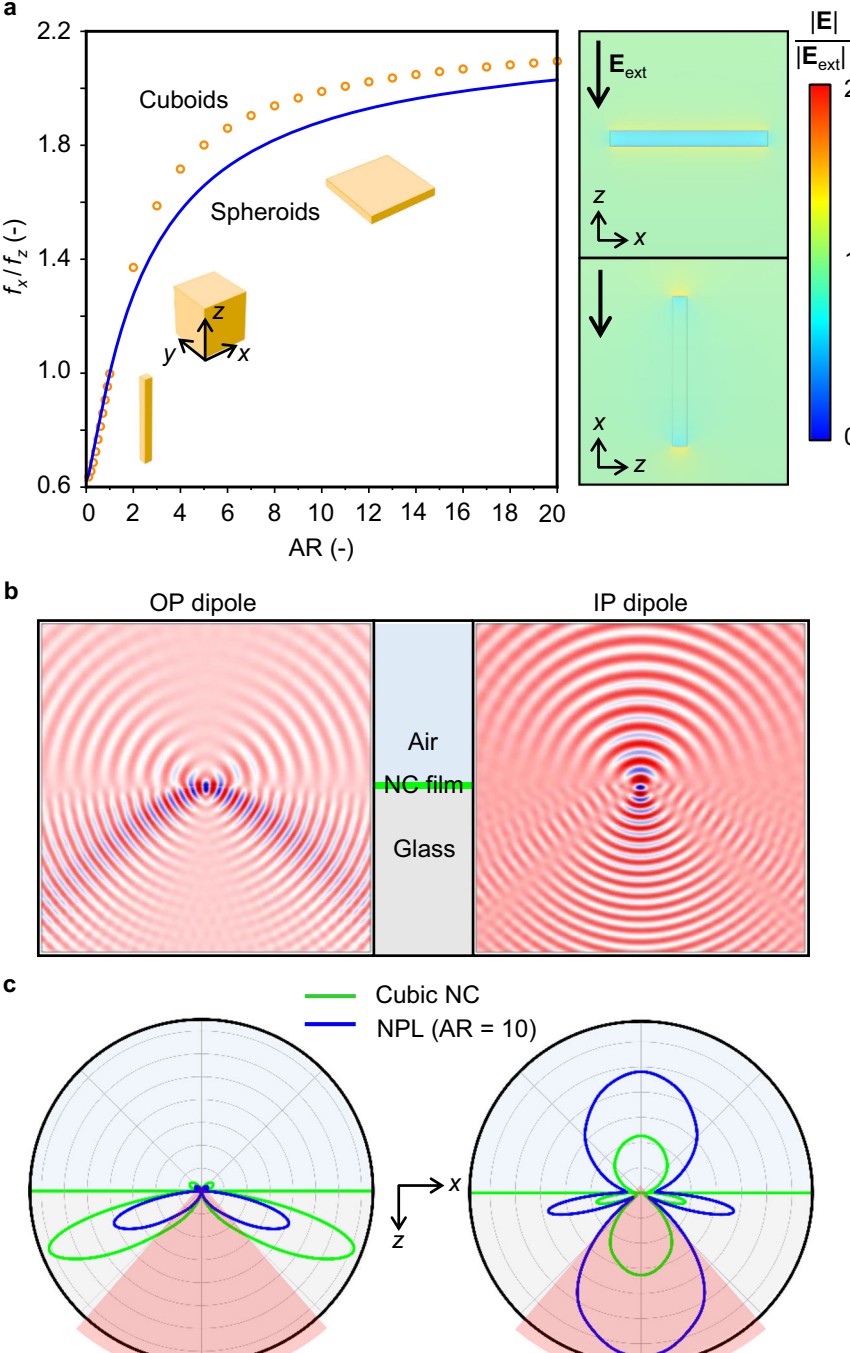

**Fig. 1 Emission polarization in anisotropically dielectric-confined NCs. a** Calculated ratio of horizontal (*x*) to vertical (*z*) local field factor, $f_x/f_z$, as a function of AR considering individual cuboidal and spheroidal perovskite NCs ($\epsilon_{NC} = 4.7$) embedded in a low-dielectric-constant ligand ($\epsilon_m = 2.129$) medium. Right panels present simulated electric field distributions near a representative square ANC of AR = 10 by applying a constant vertical (top) and horizontal (bottom) external electric field, $\mathbf{E}_{ext}$, revealing the polarization anisotropy. Calculated electric field intensity profiles (**b**) and radiation patterns (**c**) in the *x*–*z* plane for an out-of-plane (OP; left) and in-plane (IP; right) dipole placed in the center of a cubic NC and ANC of AR = 10 embedded in a dielectric emissive film sitting on the glass substrate. The dielectric anisotropy of ANC rescales the radiation from horizontal and vertical transition dipoles, thereby directing more light within the critical angle (red angular regions) for the glass-air interface (*x*–*y* plane).

The NC-shape-induced emission directionality is further illustrated in Fig. 1b, c, which compare the calculated electric fields and radiation patterns from horizontal and vertical dipoles. In these simulations, a Hertzian dipole is placed in the center of cubic NC or ANC (AR = 10) embedded in a dielectric film sitting on top of a semi-infinite glass substrate, with the NC *z* axis perpendicular to the substrate (*x*–*y* plane). The dielectric anisotropy of ANC rescales the oscillator strength for horizontal and vertical dipoles, which yield enhanced and reduced radiation power, respectively. As a result, more light can be directed within the critical angle (~41 degrees; the red angular regions in Fig. 1c) for the glass-air interface. In principle, this scenario would apply to any NC geometry, but overall, square ANCs would be the most effective for LEDs, because of the geometric similarity with the substrate.

**Directed assembly of ANCs**. The physical picture of single-dipole radiation presented in Fig. 1c is valid at device level only if individual NCs retain their radiation pattern in the assembled thin films. In practice, many undesirable effects, such as quantum resonance[46] and energy transfer[5] could come into play upon NC assembly. For example, strong face-to-face interactions between CdSe NPLs in the drop-casted film yield the "edge-on" NPL orientation with respect to the substrate, resulting in isotropic radiation pattern[36]. The emergence of LHP NCs opens an avenue to address the longstanding challenge. Indeed, the LHP dielectric response is high at low frequencies ($\epsilon > 20$), due to the lattice softness that generates strong phonon and molecular contributions, but drops rapidly in the visible regime ($\epsilon \sim 5$)[47]. As a result, the electrostatic interactions between neighboring NCs are significantly screened, largely alleviating the undesirable effects. This is reflected by our recent observations that an ultrathin organic spacer of ~0.67 nm is sufficient to decouple neighboring ANCs in their superlattices[45].

We synthesized mixed-cation perovskite ANCs with formula of $FA_{0.5}MA_{0.5}PbBr_3$, where FA = formamidinium, $CH_3(NH_2)_2^+$, and MA = methylammonium, $CH_3NH_3^+$, using the modified ligand-assisted re-precipitation method at room temperature (details see Methods)[48,49]. The ANC aspect ratio was tuned and optimized by the ligand concentration and hydrophobicity[35]. Specifically, the NC ARs can be increased by reducing the ligand hydrophobicity that stabilizes the small crystals, and increasing ligand concentration could access to few-monolayer-thin ANCs with high degrees of quantum confinement. In addition, device considerations, including balanced carrier conductivity and proper ANC dielectric confinement, were also taken into account[49]. For example, excessively thin ANCs would compromise electrical stability and thin-film impedance, both lowering the device efficiency. After extensive experimentation, mixed n-decylamine (DA) and oleic acid (OA) was chosen as the major molecular ligand unless mention otherwise. We notice that the decyl and oleic tails have similar degree of hydrophobicity. Figure 2a presents the transmission electron microscope (TEM) image for the synthesized square ANCs, having the average lateral length and thickness of $11 \pm 2$ and $4 \pm 1$ nm, respectively. Given the TEM-observed AR value (~2.75), we predict its $\Theta_{IP}$ value to be 0.80.

Figure 2e–j compares the synchrotron grazing-incidence wide-angle and small-angle X-ray scattering (GIWAXS and GISAXS) patterns for the representative emissive thin films (thickness $t \sim 30$ nm) studied here. Direct drop casting of colloidal solution on glass yields ANC solid with nearly random orientation (Fig. 2e), exhibiting the Debye–Scherrer (DS) rings corresponding to the intracrystal perovskite structure, denoted (100), (110), and (200). Shear-induced ordering during spin coating leads to the formation of superstructure with an improved degree of alignment for the ANC OP vector (Fig. 2f). The emergence of (100) Bragg peak along the $q_z$ axis and the vanishing (110) DS ring suggest that the majority of ANCs was horizontally oriented with the OP vector perpendicular to the substrate plane.

We hypothesized that the ANC stack ordering is highly influenced by the inter-NC interactions. In order to minimize the effect of underlying substrate, we examined the spin-coated ANC films sitting on a fluorinated hole transport material, X-F6-TAPC. Remarkably, the low-surface-energy surface was found to further enhance ANC ordering by forming anisotropic nanocrystal superlattices (ANSLs). The atomic force microscope (AFM) height image (Fig. 2b, c and Supplementary Fig. 6) reveals the formation of ANSLs comprised of close-packed ANC arrays with the side faces linked to each other, having the thickness of $4.6 \pm 0.5$ nm and lateral dimension exceeding 300 nm. The extracted ANCs thickness from the AFM cross-section height

profile and TEM image are in coherence with each other. The ANSL c axis coincides with the ANC OP vector. We estimate that the ~32 nm emissive layer (EML) in device contains in average 6–8 stacking layers of ANSL. The GIWAXS pattern highlights the extension of a SL Bragg rod along the $q_z$ axis, and the SL peaks corresponding to lateral packing, denoted as $(100)_{SL}$ and $(200)_{SL}$, come out on the $q_{xy}$ axis in the GISAXS pattern (Fig. 2g, i). Hereafter, we optimized our LED device based on the ANSL-contained EMLs.

Note that we observed an even higher SL crystallinity by replacing DA with oleylamine (OLA) in our synthetic protocol (Fig. 2h), as reflected by the emergence of SL Bragg peaks on the $q_z$ axis, corresponding to $(00l)_{SL}$, where $l$ is an integer. Figure 2j compares the (100) DS ring intensity as a function of polar angle $\chi$ for the four samples considered here, revealing the improvement of horizontal ANC orientation upon ANSL formation. Nevertheless, we were not able to reach high electroluminescent (EL) efficiencies with the OLA-based ANCs, hypothetically due to the long hydrophobic tail. More discussions about photophysical properties for the emissive thin films see Supplementary Section 2.1.

**Analysis of thin-film radiation patterns**. The thin-film stack (air/EML/X-F6-TAPC/glass) was mounted on a hemicylindrical glass prism using a refractive index matching optical liquid, followed by performing the polarization- and angle-dependent PL spectroscopy[50] that differentiates between the s-polarized (s-pol) emission from the transverse-electric y dipoles and the p-polarized (p-pol) emission from the transverse-magnetic x and z dipoles. The film thickness, $t$, and refractive index, $n$, for each dielectric layer was carefully characterized by ellipsometry (Supplementary Section 2.2). The generated radiation pattern resolves the PL intensity, $I$, on the substrate plane (x–y) projection of emission wave vector $k$, $k_x$, and $k_y$, which inform the effective TDM orientation in the EML within the k-space domain, $k/k_0 < n_{glass}$, where $k_0$ is the wave vector in air and $n_{glass} = 1.52$ is the refractive index of glass. It follows that $k/k_0 = 1$ corresponds to the critical angle of total internal reflection at the glass/air interface. Since the p-pol emission comes from both horizontal and vertical dipoles, optical simulations were carried out using Setfos program to fit the p-pol profile to quantify the dipole orientation, using the thin-film horizontal dipole ratio, $\Theta_H$, as the only fitting parameter [25].

Figure 3a presents the experimentally measured p-pol PL intensity as a function of viewing angle, $\phi$, together with the calculated profiles for $\Theta_H$ values of 0.67, 0.72, and 0.91, corresponding to isotropic, best-fitted, and AR $\rightarrow \infty$ models, respectively. Indeed, as revealed in Fig. 1c, near the critical angle for the glass-air interface, $\phi \sim 41°$, the emission from a horizontal dipole nearly vanishes, while a vertical dipole strongly couples into the substrate, so the emission gets intensified. As a result, a lower $\Theta_H$ would lead to a shallower minimum at $\phi \sim 41°$ and a higher fraction of light coupled into the substrate mode, $\phi > 41°$. Figure 3b compares the experimentally measured and theory-fitted k-space radiation patterns, showing excellent agreement.

We attribute the enhanced ratio of horizontal dipole to the formation of ANSL that horizontally orient individual ANCs to a great degree. Remarkably, we have carried out more angle-dependent PL measurement considering nine different ANSL thin film samples deposited on X-F6-TAPC. The statistical analysis yields consistent values of $\Theta_H = 0.73 \pm 0.016$, with the maximum of up to 0.75. Theoretical analysis taking into account the effect of NC packing disorder (Fig. S2) suggests that the orientation of ANCs in EML substantially affects $\Theta_H$. For the ANC aspect ratio value considered here, the theoretically predicted thin-film $\Theta_H$ ranges from 0.80 (order parameter $S = -0.5$) to 0.60 ($S = 1.0$).

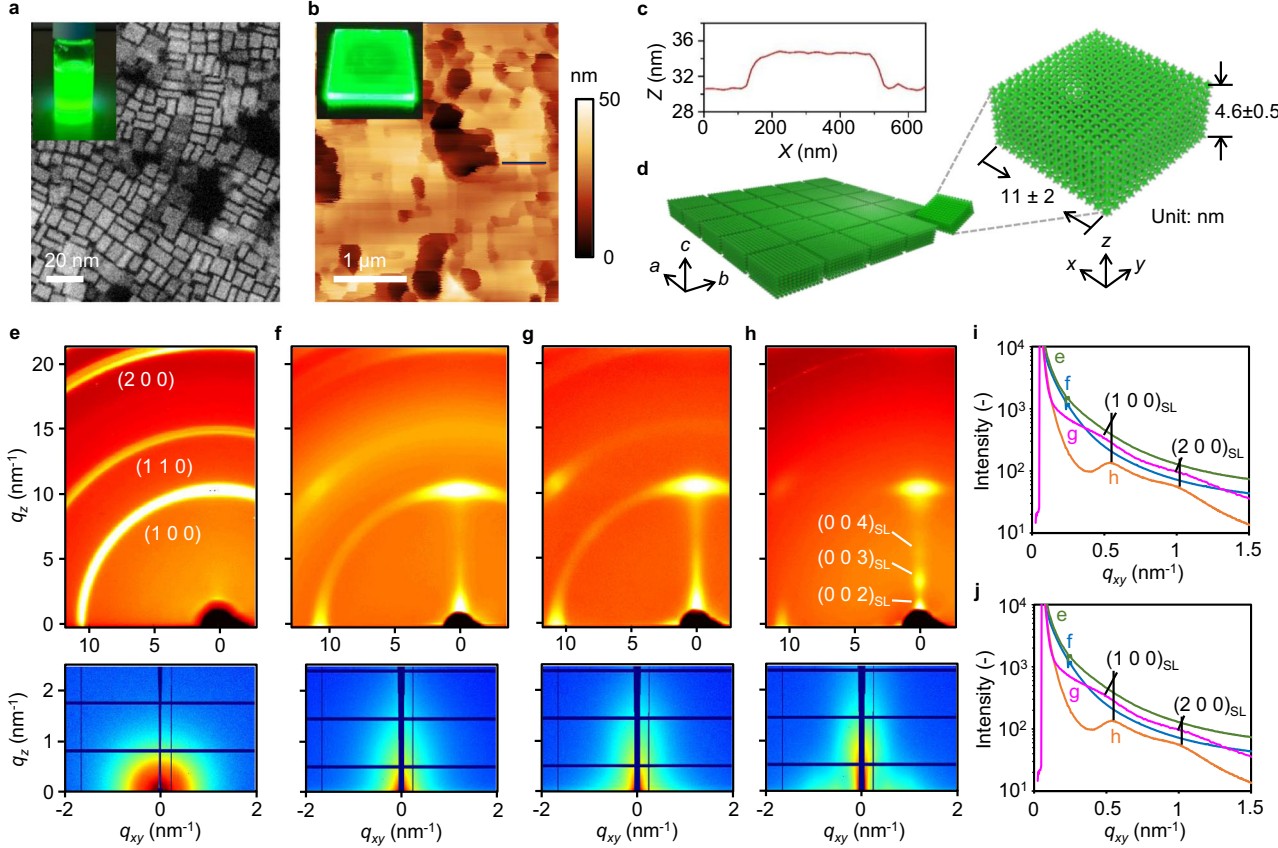

**Fig. 2 Self-assembly engineering for controlling the formation of ANC superlattices. a** TEM image and photograph of colloidal solution exposed to UV light for the synthesized ANCs. **b** Representative AFM height image and photograph exposed to UV light for the fabricated ANC superlattice film in (**g**). **c** Cross-sectional height profile corresponding to the black line in (**b**), revealing lateral assembly of ANCs in the *xy* (substrate) plane. **d** Schematics of ANC ANSL showing the superlattice *ab* plane coincides with the *xy* plane and the SL *c* axis in parallel to the *z* vector. GIWAXS (top) and GISAXS (bottom) patterns for the drop-casted film (**e**), spin-coated film on poly-TPD (**f**), spin-coated film on low-surface-energy X-F6-TAPC (**g**), and spin-coated film on X-F6-TAPC comprising NPLs synthesized using OLA ligands (**h**). The Miller index $(a\,b\,c)_{SL}$ refers to a superlattice plane. **i** $q_{xy}$ cuts for the GISAXS patterns in (**e–h**). **j** Comparison of orientation distribution function for the emissive films considered here using the normalized scattering intensity of the (100) DS ring with respect to $\chi$, revealing an enhanced horizontal orientation upon ANSL formation.

Accordingly, the single-dipole emitter model for individual ANC (Fig. 1) is already adequate to describe the collective thin-film radiation behavior.

From a fundamental point of view, the combined theoretical and experimental analysis presented here highlights the importance of NC shape to the light outcoupling efficiency in QD LEDs, which has long been ignored. We further compare the calculated far-field emission patterns (FEPs) generated by EMLs in our optimized LED stack (see details in next section) for $\Theta_H$ values of 0.44, 0.67, 0.72, and 0.91 (Fig. 3c). The $\Theta = 0.44$ model corresponds to emission in asymptotic vertical rod NC with AR → 0, the lower theoretical limit for NC solids (Supplementary Section 1.2). Regardless of $\Theta_H$ values, the calculated FEPs exhibit similarity solutions of the Lambertian function, and as expected, the radiation power increases with $\Theta_H$, thereby enhancing $\eta_{out}$ at device level. Accordingly, under the assumption that the thin-film $\Theta_H$ is equal to $\Theta_{IP}$ in individual NC, Fig. 3d presents the calculated $\eta_{out}$ as functions of AR and $\tilde{\epsilon}$. Clearly, as illustrated in Fig. 1, increasing dielectric contrast and aspect ratio would result in stronger emission polarization, effectively enhancing emission from the IP dipoles. Given a realistic range for the dielectric contrast in most semiconductor QDs, $\tilde{\epsilon} < 6$, we predict that highest attainable $\eta_{out}$ for QD solid-based device is ~40%, double the efficiency from an isotropic emitter. As for the LHP NPLs considered here (AR ~ 2.75 and $\tilde{\epsilon}$ ~ 2.12), the theoretical $\eta_{out}$ is ~32%.

**LED fabrication and characterization.** The spin-coated ANC thin films were directly employed as EMLs to examine the EL performance. We first optimized the electron transport layer (EML) material and process based on the device architecture of ITO/PEDOT:PSS/EML/ETL/LiF/Al (for full compound names see Methods). The control set of devices that used standard TPBi as ETL exhibited relatively modest peak external quantum efficiency, $\eta_{ext}$, of 5.87% and current efficiency, $\eta_{CE}$, of 24.74 cd A$^{-1}$. By replacing TPBi with 3TPYMB, the peak $\eta_{ext}$ ($\eta_{CE}$) was enhanced to 10.6% (46.6 cd A$^{-1}$), with the turn-on voltage, $V_{on}$, down to 2.75 V. Other ETL materials were also tested but did not give better performance (see Supplementary Table 3). We attribute the observed efficiency enhancement to its low electron injection barrier and relatively low refractive index, $n = 1.65$, which increases $\eta_{out}$ by reducing the substrate-mode and surface-plasmon-polariton losses [51–53].

We next optimized the hole transport layer (HTL) material and process based on the device architecture of ITO/PEDOT:PSS/HTL/EML/ETL/Liq/Al. Note that colloidal ANCs were directly spin-coated onto HTL, so the choice of HTL directly influences the behavior of ANC assembly. Numerous HTL materials were examined, while here we specifically compare three cases, without HTL, poly-TPD, and X-F6-TAPC, in which the last induces the formation of ANSLs (see Fig. 2). As for the X-F6-TAPC devices, after extensive device optimization, the following

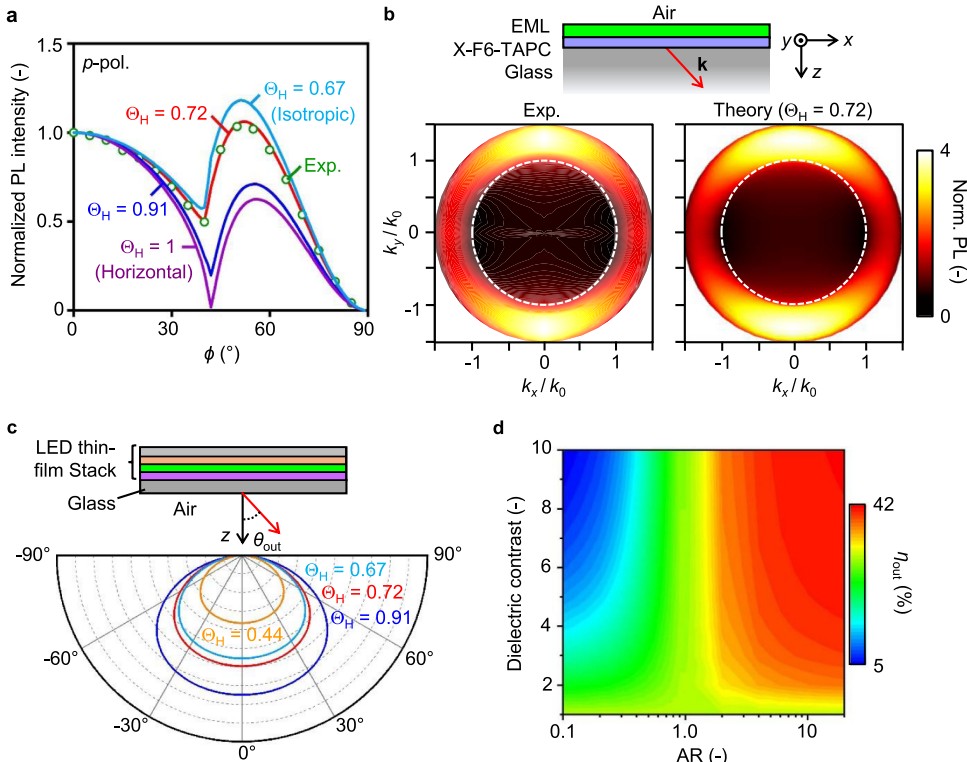

**Fig. 3 Analysis of EML radiation patterns and outcoupling efficiency calculations. a** Experimentally characterized (Exp.) and optical theory-calculated thin-film $p$-pol PL intensity as a function of viewing angle $\phi$, using the thin-film horizontal dipole ratio $\Theta_{\mathrm{H}}$ as the control parameter. **b** Experimentally characterized (left) and theory-fitted (right; $\Theta_{\mathrm{H}} = 0.72$) thin-film $k$-space radiation patterns considering the dielectric stack of EML/X-F6-TAPC/Glass. **c** Calculated far-field emission patterns (FEPs) generated by EMLs in our optimized LED stack for $\Theta_{\mathrm{H}}$ values of 0.44, 0.67, 0.72, and 0.91, revealing a stronger radiation power outcoupled to air by increasing EML $\Theta_{\mathrm{H}}$. **d** Calculated light outcoupling efficiency $\eta_{\mathrm{out}}$ as functions of the NC aspect ratio AR and dielectric contrast $\widetilde{\epsilon}$, under the assumption of thin-film $\Theta_{\mathrm{H}}$ equal to $\Theta_{\mathrm{IP}}$ in individual NC.

device architecture was developed: ITO ($120 \pm 15$ nm)/PEDOT:PSS ($30 \pm 3$ nm)/X-F6-TAPC ($18 \pm 2$ nm)/EML ($30 \pm 2$ nm)/3TPYMB (50 nm)/Liq (3 nm)/Al (70 nm). The device cross-sectional TEM image and energy diagram are shown in Fig. 4a, b. Figure 4c, d presents the current density, $J$, and luminance, $L$, as a function of voltage, $V$, for the three systems considered here. Angle-dependent EL measurement revealed that the radiation patterns are excellently Lambertian (Supplementary Fig. 28). The full angular EL distribution allow us to calculate $\eta_{\mathrm{ext}}$ as a function of $L$ (Fig. 4e). As compared to the HTL-free devices, the insertion of thin ($19 \pm 2$ nm) poly-TPD layer between ETL and PEDOT:PSS greatly enhanced the driving current and luminance, but the efficiencies slightly dropped.

In the X-F6-TAPC devices, although the low-hole-mobility nature slightly impedes hole transport[54], the peak $\eta_{\mathrm{ext}}$ is impressively improved up to 24.96% (Supplementary Fig. 39), which to our knowledge, represents the highest $\eta_{\mathrm{ext}}$ ever reported in solution-processed QD LEDs based on all semiconductor materials[4,7–9,55–61]. Supplementary Table 4 summarizes the EL performance of perovskite LEDs having LHP emissive layers with preferentially horizontal-oriented TDMs. Note that the high-efficiency range is wide, with $\eta_{\mathrm{ext}} > 20\%$ between 30 and 1500 cd m$^{-2}$ (Fig. 3e), nicely covering the brightness range of interest in commercial displays. A degree of efficiency roll-off ($\eta_{\mathrm{ext}}$ (1.6 cd m$^{-2}$) = 10.1% and $\eta_{\mathrm{ext}}$ (5009 cd m$^{-2}$) = 10.9%), nevertheless, was still observed. Moreover, the peak $\eta_{\mathrm{CE}}$ and power efficiency, $\eta_{\mathrm{PE}}$, reach 103.4 cd A$^{-1}$ and 92.8 lm W$^{-1}$ (Supplementary Fig. 39), respectively, comparable to the state-of-the-art OLEDs. We notice that the high efficiency values were obtained without using any post-synthetic defects passivating agents. The statistic histogram of 65 devices, exhibits an average

$\eta_{\mathrm{ext}}$ of 17.89% with a standard deviation of 2.62 (Fig. 4f). A possible reason resulting in a broad $\eta_{\mathrm{ext}}$ distribution is the variation of NC concentration from different synthetic batches. Although we had tried to reach the same degree of optical density for each batch of ANC colloidal dispersion used for depositing EML by evaporating/adding solvent, the resulting EML thickness remains less controllable, as compared to the evaporated organic or solution-processed bulk LHP films. The EL emission maximum and full width at half maximum (fwhm) are $528 \pm 1$ and $22 \pm 2$ nm, respectively, yielding high-color-purity green chromaticity fulfilling Rec. 2020 color gamut.

In addition to the formation of ANSL that preserves the preferentially in-plane dipole orientation in individual square ANCs, as discussed in Figs. 1–3, the following factors also contribute to the enhanced external quantum efficiency, including (i) near-unity thin-film $\eta_{\mathrm{PL}}$, (ii) cascade highest-occupied-molecular-orbital energy levels that facilitate hole injection[53,54], (iii) high lowest-unoccupied-molecular-orbital level of X-F6-TAPC that effectively confines injected electrons within the EML[53,54,62], and (iv) low refractive indices for ETL, HTL, and EML that boost $\eta_{\mathrm{out}}$ (details see Supplementary Section 2).

We also investigated operational lifetime for our X-F6-TAPC-based devices, as compared with the control device (w/o HTL). The time for the luminance to decay to 50% of the initial luminance, LT$_{50}$, under a constant current density of 0.5 mA cm$^{-2}$, corresponding to the initial luminance, $L_0$, of 276 and 125 cd m$^{-2}$ for the optimized and control devices, respectively. The optimized device exhibits LT$_{50}$ of 138 min, threefold higher than that for control device (details see Supplementary Section 3). The EL spectra remained consistent throughout the stability tests (Supplementary Fig. 41). A large-area (225 mm$^2$) device was also demonstrated

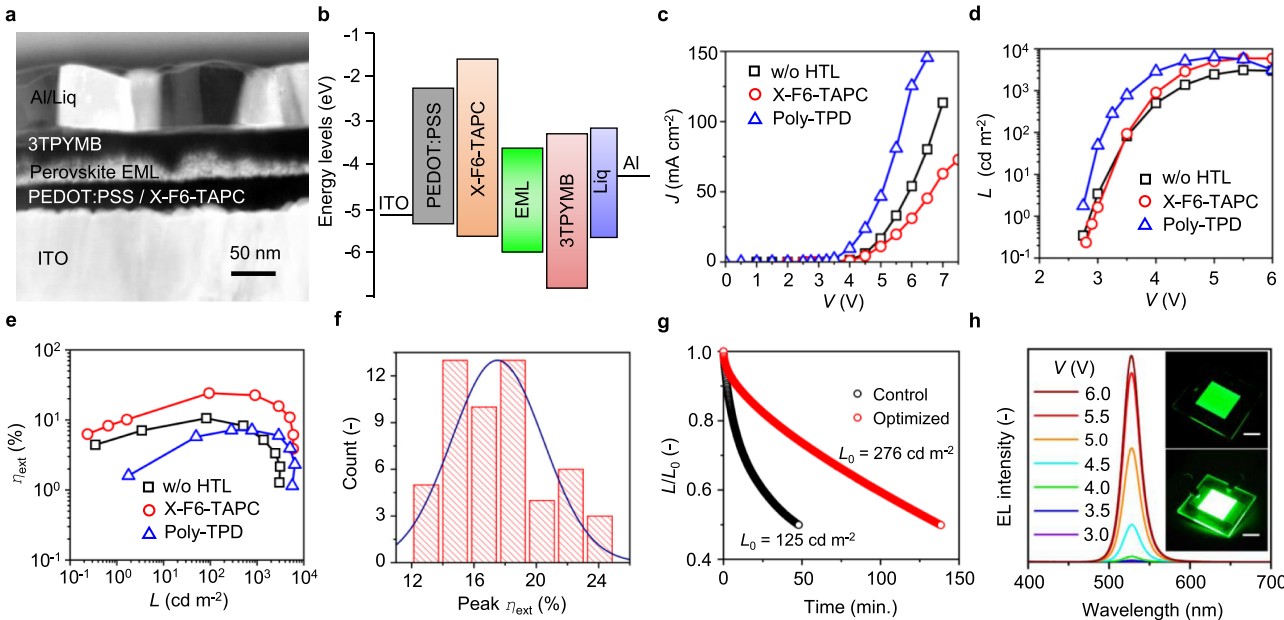

**Fig. 4 Device characteristics of ANSL-based LEDs.** Representative device cross-section TEM image (**a**) and energy diagram (**b**) of our optimized device architecture, using the EML comprised of ANSLs of ANCs. Optimized LED current density $J$ (**c**), luminance $L$ (**d**), and external quantum efficiency $\eta_{ext}$ as a function of luminance $L$ for the devices of (i) without (w/o) HTL, (ii) X-F6-TAPC HTL, and (iii) poly-TPD HTL (**e**). **f** Statistical distribution of peak $\eta_{ext}$ out of 65 LED devices. **g** Relative luminance as a function of time under continuous electrical stress at a constant current density of 0.5 mA cm$^{-2}$, corresponding to initial luminance $L_0$ of 276 and 125 cd m$^{-2}$ for the optimized and control devices, respectively. **h** EL spectra and photographs for our large-area LEDs.

without significant compromise of device performance (Fig. 4h and Supplementary Fig. 43).

Here, we have theoretically and experimentally demonstrated that the ANSL of high-aspect-ratio LHP ANCs preserve the effective in-plane dipole orientation in individual nanocrystal, thereby boosting the intrinsic light outcoupling efficiency in LEDs. Our findings lay the foundation for future QD shape and assembly engineering. In combination with the emerging defect passivating techniques, we anticipate that the performance gap between QD LEDs and OLEDs will be closed in the near future.

## Methods

**Materials for LHP NCs synthesis.** Oleic acid (OLAc, 90% technical grade, Aldrich), n-decylamine (99%, Acros Organics), dodecylamine (98%, Acros Organics), oleylamine (~90%, Acros Organics), methyl ammonium bromide (MABr, 99%, Sigma-Aldrich), formamidinium bromide (FABr, 98%, Sigma-Aldrich), lead (II) bromide (PbBr$_2$, 98%, Acros Organics), toluene (99.8%, Fisher Chemical), N,N-dimethylformamide (DMF, >99.8%, Aldrich), Ethanol (EtOH, absolute for analysis, Merck). All chemicals listed above were used without any further treatment.

**Materials for perovskite LEDs.** Patterned indium tin oxide (ITO) coated glass substrates with a sheet resistance of 15 Ω/□ and a size of 3 × 3 cm$^2$ were acquired from Lumtech Corp. The hole injection material poly(3,4- ethylene-dioxythiophene) -poly(styrene sulfonate) (PEDOT: PSS) was procured from Heraeus (Clevios AI 4083). The electron transporting materials, 2,2′,2″-(1,3,5-benzinetriyl)-tris(1-phenyl-1-H-ben-zimidazole) (TPBi; 99%), tris(2,4,6-trimethyl-3-(pyridin-3-yl)phenyl)borane (3TPYMB; 99%) and 2,4,6-tris[3-(diphenylphosphinyl)phenyl]-1,3,5-triazine (PO-T2T; 99%), polymeric hole transporting material, poly(N,N′-bis(4-butylphenyl)-N,N′-bisphe-nylbenzidine) (Poly-TPD), N,N′-(4,4′-(cyclohexane-1,1-diyl)bis(4,1-phenylene))bis(N-(4-(6-(2-ethyloxetan-2-yloxy)hexyl)-phenyl)-3,4,5-trifluoro aniline) (X-F6-TAPC), and electron injection material, Liq (99.5%), were also procured from Lumtech Corp. The electron transporting material, 4,6-bis(3,5-di(pyridin-3-yl)phenyl)-2-methylpyrimidine (B3PYMPM; 99.5%), was purchased from Flask.co.jp. The electron injection material lithium fluoride (LiF; 99.98%) is purchased from Acros Organics. Aluminum (Al) pellets (99.999%) were purchased from Kurt J. Lesker Co. Ltd. All materials were used without any further purification.

**Synthesis of LHP ANCs.** The colloidal dispersion of ANCs of FA$_{0.5}$MA$_{0.5}$PbBr$_3$ were synthesized using modified synthetic protocol from our earlier report[63]. The reaction was carried out under ambient conditions with a relative humidity of 50 ± 20% and a room temperature of 25 °C. Firstly, a 12.5 ml nonpolar toluene was taken into a round bottom (RB) flask and start vigorous stirring. The 0.625 ml OLAc and 0.03 ml of DA and 0.035 ml of dodecylamine were consequently added in the RB flask. Before mixing in the RB flask, the FABr and MABr (0.6 M) precursors were distinctly dissolved in ethanol, while PbBr$_2$ (0.6 M) was dissolved in polar N,N-dimethylformamide (DMF) solvent. Moreover, the FABr and MABr precursor solutions were premixed in the ratio of 1:1 before mixing in the RB flask. Subsequently, the precursor solutions of PbBr$_2$ (0.625 ml) and a mixture of FABr and MABr (0.625 ml) were then added dropwise in the RB flask consisting of a non-polar toluene solution of long-chain organic surfactants, DA as a long-chain ligand, and OLAc as a stabilizer. An instantaneous colloidal crystallization is triggered due to poor solubility of perovskite precursors in the nonpolar toluene. Upon centrifugation at 6797 g for 10 min, the reaction mixture is separated in supernatant and precipitate. The resultant supernatant is discarded and the pre-cipitate containing perovskite nanocrystals is redispersed in 2.0 ml fresh toluene. Finally, the solution was centrifuged again, and filtered through a 0.2 μm pore size Teflon filter to obtain the colloidal dispersion of ANCs. The concentration of ANCs is adjusted by adding the toluene to the colloidal dispersion.

**Synchrotron grazing-incidence wide-angle and small-angle X-ray scattering (GIWAXS) and (GISAXS).** Synchrotron GIWAXS and GISAXS was conducted at the Taiwan Photon Source beamline BL13A at the National Synchrotron Radiation Research Center of Taiwan. The incidence angle and beam energy of the X-ray were 0.12 and 12.16 keV, corresponding to a wavelength of 1.02143 Å. The MAR165 CCD with a 2D area detector was used to collect all of the GISAXS images in reflection mode.

**Absorption spectrum measurement.** Absorption spectra of colloidal ANCs were measured using a JASCO V670 UV–VIS–NIR Spectrophotometer.

**Absolute PL quantum efficiency ($\eta_{PL}$) and PL measurement.** The absolute $\eta_{PL}$ was determined using the Quantaurus QY (C11347-11) from Hamamatsu equipped with 150 W Xenon light source and a 3.3 inches integrating sphere, which is coated with highly reflective Spectralon. The $\eta_{PL}$ of samples were characterized through varying the excitation wavelengths ($\lambda_{ext}$) between 350 and 500 nm. The excitation power dependent $\eta_{PL}$ of samples were measured by using the Quantaurus-QY Plus UV–NIR absolute PL quantum yield spectrometer (C13534-12) equipped with a power tunable excitation source Lightningcure LC8. The samples were excited at an excitation wavelength of 405 nm by using a short band pass filter that resulted in an absolute excitation power of 12 W m$^{-2}$, which is tuned between 1 and 100%. The PL spectrum was recorded with Hamamatsu CCD PMA-12 spectrometer (wavelength resolution <2 nm).

**Time resolved photoluminescence (TRPL) spectroscopy**. TRPL spectra were characterized using a Hamamatsu Quantaurus-Tau (Q-Tau) fluorescence lifetime spectrometer (C11367-31) equipped with a photon counting measurement system. The ANCs thin film samples were exited at 470 nm pulsed emission with a repetition rate of 200 kHz and 10,000 counts. The excitation wavelength of 470 nm was chosen to avoid excitation of underneath HTL, X-F6-TAPC, and Poly-TPD. The PL decay curves of LHP ANCs on different surfaces were fitted using a biexponential decay model. The excitation power dependent PL lifetime measurements were performed using a time correlated single photon counting (TCSPC) setup, equipped with an SPC-130-EM counting module (Becker & Hickl GmbH) and an IDQ-ID-100-20-ULN avalanche photodiode (Quantique) for recording the decay traces. The samples were excited by 355 nm pulse laser with a maximum pulse intensity of 8.5 nJ cm$^{-2}$ triggering a TCSPC counting module through an electronic delay generator (DG535 from Stanford Research Systems). The pulse laser intensity tuned between 0.1 and 100%.

**Focused ion beam/scanning electron microscopy (FIB/SEM)**. Cross-sectional lamellae of perovskite LED devices were fabricated using a Thermo Scientific Helios 5 UX FIB/SEM at ScopeM, the Scientific Center for Optical and Electron Microscopy at ETH Zürich. All lamellae were prepared in the cross-section to device stack. Two amorphous carbon protection capping layers were deposited to avoid the electron-beam damage during lamellae fabrication and cross-section TEM imaging. The first capping layer was deposited using the FIB at an acceleration voltage of 2 kV and a current of 6.4 nA with a nominal thickness of 500 nm. Thereafter, the automated TEM lamella preparation (AutoTEM 5) was started to deposit the second carbon capping by the FIB at 30 kV and 0.26 nA. Subsequently, the coarse trenches were milled at 20 nA and polished at 9.4 nA. After the automated transfer to the TEM grid each lamella was thinned using a sequence of decreasing currents at 30 kV, followed by one polishing step at 5 kV.

**High-angle annular dark-field scanning transmission electron microscopy (HAADF-STEM) and high-resolution transmission electron microscopy (HR-TEM)**. HAADF-STEM images of LHP ANCs were acquired in cryogenic conditions using Hitachi HD 2700 CS equipped with cryo-holder (liquid nitrogen) with high beam acceleration voltage. The HAADF-STEM cross-section image of perovskite LED device was acquired from the lamellae using a FEI Talos F200X operating at 200 kV with the HAADF-STEM and HR-TEM modes. The Energy dispersive X-ray spectroscopy (EDS) mapping of various layers in the device cross-section was acquired by using quadrant EDS X-ray detector (Super-X, Thermo Fisher Scientific, the Netherland).

**Atomic force microscopy (AFM)**. AFM topographic images of LHP ANCs thin film was captured using the Bruker BioScope Resolve AFM using PeakForce Tapping mode in combination with ScanAsyst automatic parameter adjustment functionality. The topographic AFM images were captured using a 1 nm ultrasharp probe with a peak force amplitude of 50 nm, peak force set-point of 0.1 nN, and a scan rate of 1.4 Hz.

**Spectroscopic ellipsometry (SE)**. Colloidal solution of ANCs and hole transporting materials, X-F6-TAPC and Poly-TPD, were spin-coated on oxygen plasma clean Si wafer having a 302 nm SiO$_2$ layer. The amplitude (psi; $\psi$) and phase shift (delta; $\Delta$) plot acquired through a micro-spot Spectroscopic Ellipsometer from SENTECH SE850 at a variable incidence angle of 60°, 65°, and 70° with an incident light wavelength ranging between 350 nm and 850 nm. The optical constants of all thin films were calculated by fitting the raw data $\psi$ and $\Delta$ plot with the SENTECH SpectraRay2 (SR2) or Fluxim Setfos 4.6 software program. Firstly, the non-absorbing region, 600–850 nm, of optical index data was fitted with the Sellmeier dispersion equation.

$$n_{SL}^2(\lambda) = \sum_{i=1}^{3} \frac{A_i \lambda^2}{\lambda^2 - B_i} \qquad (2)$$

Generally, the all six Sellmeier parameters (A1, A2, A3 and B1, B2, and B3) as well as film thickness ($t$) were determined using spin-coated film using SpectraRay2 analysis software. Furthermore, the analysis was completed when Tauc–Lorentz (TL) model was applied in order to account for optical absorption. An accurate fit was obtained with two TL oscillators in the range of 400–850 nm. Moreover, it is important to note that SR2 uses different definition of $\Delta$ than Setfos 4.6, in which one had to subtract 180° from the values obtained experimentally (SR2) in order to process them in Setfos 4.6.

**Orientation of PL transition dipole moment and momentum-resolved (k-space) photoluminescence**. The angle dependent PL spectra of ANCs film were characterized using a commercial Phelos instrument (Fluxim Inc.) equipped with a CCD spectrometer and a polarizer with a hemisphere glass lens. This feature allows the extraction of photons with a normalized wave vector $k/k_0 > 1$, usually lost in substrate modes. First, the colloidal solutions of perovskite samples were spin-casted on the glass substrates. Then, the substrate was placed on the top of the hemispherical glass lens using a refractive index matching liquid. The latter ensures

a lack of air in the substrate-lens interface. Thereafter, an LED head emitting light at 365 nm was mounted on top and a 3 × 5 mm$^2$ spot of the sample was excited. A typical measurement procedure consisted of simultaneous sweeping both polarization ($\theta$) and viewing angles ($\varphi$). The PL emission is measured at different angles by varying the viewing angle from 0° to 85°, in the steps of 5°, whereas the polarizer angle was varied between 0° to 90° with the step size of 10°, where $\theta = 0°$ corresponds to p- and $\theta = 90°$ to s-polarization.

All angular PL emission patterns were converted to k-space. For each polarization angle, the relation $I(\varphi)$ vs. $\varphi$, which is obtained experimentally, can be transformed into $I(k/k_0)$ vs. $k/k_0$ using the following relations[34]:

$$\frac{k_x}{k_0} = n_{sub} * \sin\varphi * \cos\theta \qquad (3)$$

$$\frac{k_y}{k_o} = n_{sub} * \sin\varphi * \sin\theta \qquad (4)$$

$$I(k/k_0) = \frac{I(\varphi, \theta)}{\cos\varphi} * C \qquad (5)$$

where $k/k_0$ and $n_{sub}$ represent the normalized wave vector and substrate refractive index ($n$), respectively. $C$ equals to $\sqrt{\varepsilon} * \omega * c$, with $\varepsilon$ being the permeability of the glass substrate at the emission frequency $\omega$. We observed a narrow PL emission in the ANC samples, which remained unchanged with varying the viewing angle as well as polarization angle. Therefore, we assumed the emission width to be a constant. Furthermore, in most of the cases shown in this work, all intensities were normalized and therefore a quantitative determination of $C$ was not necessary. As a result, all transformed data could be plotted as 2D contour plots, thus generating k-space radiation patterns. These were then compared to theoretically predicted ones.

The experimental data was evaluated with the computation software Setfos 4.6 provided by Fluxim Inc. A precise film thickness ($t$) as well as a relation between emitter $n$ value and incident light wavelength ($\lambda$) were obtained by means of SE and used as input parameters for the optical model. We assumed the dispersion of $n$ value with respect to $\lambda$ is considered. Setfos allows to compute the coherent light propagation in individual optical layers/cavities as well as across multilayer stacks by considering the respective polarization-dependent Fresnel reflection and transmission coefficients at each of the interfaces[64,65]. Light generation inside an active layer is described as power radiated by spatially distributed electrical dipoles (dipole moment $\vec{p}$[66–68]).

Here, Setfos package was used to simulate the angle-dependent s- and p-polarized PL intensity, $I_p(\varphi)$ and $I_s(\varphi)$, from the ANCs thin film sample attached to the hemispherical glass lens, for the given input parameters $t$, $n_{SL}(\lambda)$. The only fitting parameter, namely the emission dipole orientation

$$R_{IP} = \frac{\sum p_x^2 + p_y^2}{\sum \vec{p}^2} \qquad (6)$$

representing the fraction of in-plane oriented dipoles, was determined by fitting of p-polarized emission $I_p$ as a function of $\varphi$. Based on the computed $R_{IP}$ and other input parameters, the s-polarized emission $I_s$ was calculated afterward and compared to experimental data.

For the sake of comparability with the measurement, the simulated emission patterns for polarization angles $0° < \theta < 90°$ were calculated as a superposition of $I_p(\varphi)$ and $I_s(\varphi)$,

$$I(\theta, \varphi) = I_p(\varphi) * (\cos\theta)^2 + I_s(\varphi) * (\sin\theta)^2. \qquad (7)$$

Finally, all computed $I(\theta, \varphi)$ were treated analogously to experimental data, using Eqs. (2)–(4).

**Optical simulations for light out-coupling**. The light out-coupling in the optimized perovskite LED device was computed using a commercial software program Setfos 4.6 from Fluxim Inc. The device with a stacking layer sequence of ITO (120 nm)/PEDOT:PSS (35 nm)/X-F6-TAPC (x nm)/ANCs film (32 nm)/3TPYMB (y nm)/Liq (3 nm)/Al (70 nm) was utilized for computations. The x and y values were varied between 5 and 200 nm to compute the out-coupling efficiency ($\eta_{out}$). The experimental characteristics, such as, refractive index (n), individual thickness of each layer, photoluminescence spectra, and orientation of emission transition dipole moment ($\Theta$) of ANCs were independently characterized to simulate the $\eta_{out}$ by varying the thicknesses of carrier transporting layers, X-F6-TAPC and 3TPYMB. Moreover, the Gaussian distribution of recombination profile was assumed for mode analysis to obtain the various mode of losses, including substrate, absorption, waveguide, and evanescent losses.

Setfos computes the propagation of electromagnetic plane waves through a stack of individual optical layers by considering the respective polarization-dependent Fresnel reflection and transmission coefficients at each of the interfaces through a transfer-matrix method[65]. The electric field of the emitting dipole sources **E** is given by

$$\mathbf{E} = k^2 \Pi + \nabla(\nabla \Pi) \qquad (8)$$

where $\pi$ is the Hertz vector which can be written in terms of a Sommerfeld

expansion[69] in cylindrical coordinates $(r,\phi,z)$.

$$\Pi = \frac{ip}{4\pi\epsilon} \int_0^\infty \frac{u}{l} \exp(il|z|) J_0(ur)\,du \qquad (9)$$

where $p$ is the dipole moment, $u$ is the normalized surface-parallel wave vector and $J_0$ is the zeroth order Bessel function. In the specific case of a dipole located within a planar multilayered structure, the source term is extended by the respective Fresnel reflection coefficients. The outcoupling efficiency is then obtained by integrating the normal component of the Poynting vector within the escape cone and normalizing it by the dipole decay rate. Similarly, the power coupled in the other modes is calculated by selecting the appropriate integration intervals.

**Finite elements optical simulations**. The electric dipole fields have been calculated using the commercial finite element software COMSOL Multiphysics (Electromagnetic Waves Frequency Domain Module). An electrical point dipole source has been placed into a NC of specified AR and dielectric constant $\epsilon_s$. The NC is embedded in a thin layer of dielectric constant $\epsilon_m$ which lies on top of a semi-infinite glass substrate plane ($n = 1.52$). The radiation pattern is calculated by evaluating the magnitude of the real part of the Poynting vector on a circle boundary enclosing the source whose radius ($r > \frac{2c}{\nu}$) is selected to ensure the radiation patterns are probed in the farfield.

**Fabrication of perovskite LED devices**. Patterned ITO coated glass substrates were rinsed with Extran MA02 neutral detergent and deionized water mixture (1:3). Afterward, these substrates were sequentially sonicated in acetone and iso-propanol, each for 20 min. The substrates were then exposed to oxygen plasma for 10 min. in diener plasma cleaner using 80% lamp power. Thereafter, the aqueous PEDOT: PSS solution was spin-coated on the pre-cleaned ITO glass at a spin speed of 4000 rpm for 50 s. All substrates were then transported to nitrogen atmosphere glovebox. These substrates were annealed at 120 °C for 0.5 h in the glovebox. Then successive layers were deposited through spin-casting and thermal evaporation, respectively. A hole transporting layer, Poly-TPD or X-F6-TAPC, was deposited on the PEDOT:PSS layer with a spin rate of 2500 rpm for 40 s. Subsequently, the anisotropically confined LHP NC films were casted at a spin rate of 2500 rpm for 40 s. On the one hand, the cross-linking reaction of X-F6-TAPC layer was initiated through exposing a 365 nm UV irradiation for 20 s then cross-linked at 120 °C for 0.5 h. On the other hand, the Poly-TPD layer was then annealed at 120 °C for 0.5 h. Subsequently, a ~30 ± 2 nm LHP NCs film was also spin-casted at 2500 rpm with acceleration rate of 400 rpm for 40 s. All substrates were mounted on a substrate holder, which is then transferred into an ultrahigh vacuum evaporation chamber. Thus, a 50 nm ETL was deposited on the EML by the thermal evaporation. Lastly, a 1 nm LiF or a 3 nm Liq electron injection layer and a 70 nm Al cathode layer were also deposited in a high vacuum chamber ($1 \times 10^{-7}$ mbar) by using a shadow mask. Each substrate is patterned to realize device active area, which is 25 mm² for the small area devices and 225 mm² for the large area devices, as defined by the overlapping between bottom ITO anode and top Al cathode layers. Finally, these devices were stored in the glove box and characterized in the ambient atmosphere at a room temperature of 25 ± 5 °C and relative humidity of 50 ± 20%.

**Characterization of perovskite LED devices**. Current density–voltage–luminance ($J$–$V$–$L$) characteristics of the perovskite QD LEDs were measured using a Photo Research PR 655 SpectraScan spectrometer and Keithley 2400 source meter. The electroluminance (EL) spectra of all the devices were also recorded by using a PR 655 spectrometer. The $\eta_{ext}$ was calculated as the total number of emitted photons divided by the total number of injected electrons. Moreover, the angular distribution of EL emission of perovskite QD LEDs is characterized by the angular EL measurement system (C9920-11; Hamamatsu Photonics). The $\eta_{ext}$ characterization set-up further verified with an absolute $\eta_{ext}$ measurement system (C9920-12; Hamamatsu Photonics and Phelos instrument; Fluxim Inc.) consisting a 3.3 inches reflective Spectralon coated integrating sphere (A10094), a highly sensitive CCD Spectrometer PMA-12 Photonic multichannel analyzer (C10027-01), a fiber probe, a computer-controlled Keithley 2400 source meter unit (C13697-01), and data analyzer EL efficiency measurement software package (U6039-06), which is analogous to that described by Jeong et al.[70]. The operational lifetime of perovskite QD LED devices was measured using smart Ossila Lifetime System (E642) under a constant current. The surface temperature images of the perovskite QD LED devices were recorded using FLIR SC7500 MWBB Thermal Imaging Camera equipped with 50 mm lens. The images were recorded in continuous mode with a frame rate of 100 Hz and sub-millisecond integration time.

## Data availability
The data that support the findings of this study are available in the paper, Supplementary Information, as well as from the corresponding authors upon reasonable request.

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

## Acknowledgements

C.J.S., S.K., and T.M. are grateful for financial support from the ETH research grant (ETH-33 18-2), Swiss National Science Foundation project grant (project number: 200021-178944), and European Research Council (ERC) starting grant (N849229 - CQWLED). In addition, the technical support from the FIRST Lab in ETH Zurich is highly appreciated. The authors appreciate technical support of Dr. Reuteler Joakim and Dr. Zeng Peng from ScopeM, the Scientific Center for Optical and Electron Microscopy at ETH Zürich. We thank Dr. Sergii Yakunin for power dependent time resolved PL lifetime and PL quantum efficiency characterization of ANCs thin films. We also cherish support from Dr. Efstratios Mitridis for infrared imaging of device surface temperature.

## Author contributions

S.K., T.M., and C.J.S. conceived the idea and designed the experiments. S.K. synthesized CQWs and carried out their photophysical, optical, and morphological characterization. S.K. designed, fabricated, and characterized the perovskite LED devices. S.K. and T.M. performed optical simulations. T.M. performed most theoretical analysis. T.M. performed GISAXS under supervision of Y.C.C. F.K acquired cryo STEM images of perovskite NCs. S.K., T.M., and C.J.S. co-wrote the paper. All authors contributed to this work, commented on the paper, and agreed to the contents of the paper and supplementary materials.

## Competing interests

The authors declare no competing interests.
