## [Peer Review File · Nature Communications]

REVIEWERS' COMMENTS

Reviewer #1 (Remarks to the Author):

Comments to NCOMMS-22-06511-T “Anisotropic Nanocrystal Superlattices Overcoming Intrinsic Light Outcoupling 3 Efficiency Limit in Perovskite Quantum Dot Light-Emitting Diodes”

Comments:

We thoroughly reviewed the revised manuscript and found that all comments were cleared. This work is very important in this field because horizontally orienting the TDM of perovskite emitter is the ultimate strategy to enhance the EQE of PeLEDs over ~28%, and possibly up to 40%. We highly value this work because of the fundamental and scientific study of the optical properties of the perovskite ANSLs which have not been explored yet. The response to our Comments in the previous review are reasonable and the question have been completely cleared. Especially, the responses to the comments 1 and 4 are very detailed and important. We recommend the authors include the response to the comment 1 about the refractive index of ANSLs in the supporting information since it would be very helpful for the readers for the understanding and further experiments. We think that the scientific discovery and quality of this manuscript are sufficient for publication in *Nature Communications*, and we recommend the manuscript be accepted.

Reviewer #2 (Remarks to the Author):

I think that the manuscript can be published in the present form.

Response Letter to Reviewers

Reviewer #1:

We thoroughly reviewed the revised manuscript and found that all comments were cleared. This work is very important in this field because horizontally orienting the TDM of perovskite emitter is the ultimate strategy to enhance the EQE of PeLEDs over ~28%, and possibly up to 40%. We highly value this work because of the fundamental and scientific study of the optical properties of the perovskite ANSLs which have not been explored yet. The response to our Comments in the previous review are reasonable and the questions have been completely cleared. Especially, the responses to the comments 1 and 4 are very detailed and important. We recommend the authors include the response to the comment 1 about the refractive index of ANSLs in the supporting information since it would be very helpful for the readers for the understanding and further experiments. We think that the scientific discovery and quality of this manuscript are sufficient for publication in Nature Communications, and we recommend the manuscript be accepted.

We thank the reviewer for his support for publication of our article in the Nature Communications. We have also included the response to comment 1 in the Supporting Information.

Reviewer #2:

I think that the manuscript can be published in the present form.

We thank the reviewer for his support for publication of our article in the Nature Communications.